# Large spatial variations in the flux balance along the front of a Greenland tidewater glacier

Till J. W. Wagner[1], Fiamma Straneo[2], Clark G. Richards[3], Donald A. Slater[2], Laura A. Stevens[4], Sarah B. Das[5], and Hanumant Singh[6]

[1]Department of Physics and Physical Oceanography, University of North Carolina Wilmington, NC 28403, USA
[2]Scripps Institution of Oceanography, University of California at San Diego, La Jolla, CA 92093, USA
[3]Bedford Institute of Oceanography, Fisheries and Oceans Canada, Dartmouth, NS B2Y 4A2, Canada
[4]Lamont-Doherty Earth Observatory, Columbia University, Palisades, NY 10964, USA
[5]Department of Geology and Geophysics, Woods Hole Oceanographic Institution, Woods Hole, MA 02543, USA
[6]Department of Electrical & Computer Engineering, Northeastern University, Boston, MA 02115, USA

Correspondence: Till J. W. Wagner (wagnert@uncw.edu)

**Abstract.**

The frontal flux balance of a medium-sized tidewater glacier in western Greenland in the summer is assessed by quantifying the individual components (ice flux, retreat, calving, and submarine melting) through a combination of data and models. Ice flux and retreat are obtained from satellite data. Submarine melting is derived using a high resolution ocean model informed by near-ice observations, and calving is estimated using a record of calving events along the ice front. All terms exhibit large spatial variability along the ~ 5 km wide ice front. It is found that submarine melting accounts for much of the frontal ablation in small regions where two subglacial discharge plumes emerge at the ice front. Away from the subglacial plumes, the estimated melting accounts for a small fraction of frontal ablation. Glacier-wide, these estimates suggest that mass loss is largely controlled by calving. This result, however, is at odds with the limited presence of icebergs at this calving front - suggesting that melt rates in regions outside of the subglacial plumes may be underestimated. Finally, we argue that localized melt incisions into the glacier front can be significant drivers of calving. Our results suggest a complex interplay of melting and calving marked by high spatial variability along the glacier front.

## 1 Introduction

The retreat of Greenland's tidewater glaciers may be among the most noticeable manifestations of a changing global climate (Jensen et al., 2016; Carr et al., 2017). Tidewater glaciers act as thermodynamic buffers as well as mechanical buttresses between the ocean and the main Greenland ice sheet (Rignot and Thomas, 2002; Howat et al., 2007; Nick et al., 2009). The speed-up of the Greenland ice sheet observed since the early 2000s (Howat et al., 2008; Moon et al., 2012) has likely been caused (at least to some degree) by the thinning of the glaciers' termini (Vieli and Nick, 2011) and, in some cases, the disappearance of their floating tongues (Holland et al., 2008; Wilson et al., 2017; Hill et al., 2017). The processes that determine the flux balance at the glacier front therefore impact the ice sheet as a whole, yet a comprehensive understanding of these processes remains elusive. Increased ocean and air temperatures are expected to further increase the rates of glacier

retreat in the coming decades (Joughin et al., 2012; Nick et al., 2013), lending additional weight and urgency to the study of calving front dynamics.

For a retreating glacier, the delivery of upstream ice to the terminus is outweighed by the loss of ice due to frontal ablation. At tidewater glaciers this frontal ablation occurs predominantly through two distinct processes: submarine melting and calving,

both of which remain very difficult to constrain observationally.

Recent studies have reported ways to measure submarine melting either directly (from repeat multibeam sonar surveys; Fried et al., 2015), or indirectly (by considering the ocean heat transport toward the glacier; Jackson and Straneo, 2016). In most cases, however, melt is estimated using parameteriziations which require local ocean temperatures and water velocities (Holland and Jenkins, 1999). Constraining melt rates at glacier fronts then relies on accurate observations of the ocean properties at

these hard-to-reach ice–ocean interfaces and on finding appropriate parameterizations that translate these observations to melt rates. The continued scarcity of near-terminus data results in large uncertainties in current melt parameterizations (Straneo and Cenedese, 2015).

While melting is a continuous process, calving is discontinuous, highly complex, and influenced by a multitude of environmental factors, as well as the condition of the ice itself (Benn et al., 2007). In recent years, much effort has been dedicated to

studying the calving of tidewater glaciers (see the review by Benn et al., 2017), yet a comprehensive understanding of what processes and variables determine the frequency and magnitude of calving events remains elusive.

Oftentimes calving and melt fluxes are not considered separately, but rather as a single ablation term, in particular when derived from satellite imagery (Luckman et al., 2015). In situ ablation data remains scarce, and previous studies of explicit calving activities of Greenland's tidewater glaciers have typically been limited to visible daylight hours (see, for example, the

calving event catalogue of Åström et al., 2014), or somewhat indirect detection methods such as teleseismicity (Veitch and Nettles, 2012) and measuring calving-generated surface gravity waves (Minowa et al., 2018).

Finally, the calving and melt fluxes of glaciers are oftentimes described by single (horizontally and vertically-averaged) mean values (Rignot et al., 2016). However, both melting and calving can vary substantially along the front of a glacier, with largely unknown implications for the overall stability of a glacier front. For example, submarine melt is enhanced in the vicinity of

subglacial discharge plumes, leading to pronounced undercutting and incisions into the ice front (Fried et al., 2015). Spatially resolving these differences is challenging, and in particular spatial calving distributions are difficult to obtain.

Here we use a multifaceted dataset for a first attempt at quantifying the relative contribution of calving and melting and their spatial variability along a glacier front. The dataset consists of both in situ and remotely-sensed observations of the front of Saqqarliup Sermia, a mid-sized Greenland tidewater glacier. The dataset is unique in its detail, close proximity to the glacier

front, and in that it contains observations of all of the main physical quantities of interest. The dataset consists of (i) detailed bathymetry at the glacier front, (ii) high-resolution ice-surface elevations, (iii) InSAR-derived ice velocities at and upstream from the glacier front, (iv) a continuous 3-week calving event catalogue, (v) local hydrographic measurements that allow for estimates of melt rates, and (vi) multibeam sonar imagery of the underwater shape of the glacier front. The spatial and temporal concurrence of these observations allows us to compare and contrast the individual components that make up the frontal mass

budget of the glacier.

Specifically, we first derive ice flux and retreat using satellite data collected over the observational period. We then compute submarine melting using a numerical model that is constrained (and validated) by near-ice hydrographic observations. Next, we estimate calving as a residual of the other terms in the frontal mass budget and compare this estimate with the observed calving frequencies. Finally, we bring our findings together to assess the overall mass budget and discuss how calving may be enhanced by highly focused melt "hotspots".

## 2    Field campaigns and physical setting

Saqqarliup Glacier and the adjacent Sarqardleq Fjord were visited during two field seasons in the summers of 2012 and 2013. This site was chosen because ocean properties and bathymetry could be measured within 100 m of the ice front. Such observations are exceedingly difficult to obtain at larger glaciers which often have an ice mélange that obstructs access and where calving poses a major threat to equipment and personnel. The fjord is a tributary to the Ilulissat Icefjord, with the north-west facing front of the glacier (Fig 1) located 30 km south-east of Ilulissat Icefjord. At the glacier front, the fjord is about 5 km wide and the terminus is mostly, if not completely, grounded.

Since 2004, the main north-eastern part of the terminus has been retreating more rapidly than the south-western section, which now juts out by almost 1 km from the rest of the glacier front (Fig S2). This part of the glacier, which we refer to as the "promontory" (Fig 1), is grounded in shallow bathymetry and features tall ice cliffs (40–50 m above mean sea level, see Section 2.2). Overall, the glacier advanced slightly between 1975 and the early 1990s, but experienced an accelerating retreat from the mid-1990s until 2016 (Fig S2;  Stevens et al., 2016). The front position has been relatively stable from 2016 to 2018.

The 2012 survey, described by Stevens et al. (2016), revealed the presence of two main subglacial discharge plumes along the glacier front which, in turn, drained the two dominant catchment basins. The plume entering the fjord at the eastern edge of the promontory (Fig 1) has an order of magnitude greater drainage and can result in an outcropping surface pool (Mankoff et al., 2016). We refer to this as the "main plume". While this plume appears to be an annually recurring feature, its discharge is likely amplified episodically by the cyclical drainage of the ice-dammed lake Tininnilik located to the south-west of the promontory (Kjeldsen et al., 2017). We note that the dramatic retreat of the glacier front in 2015 coincided with a major drainage event of lake Tininnilik (Kjeldsen et al., 2017). The second recurring plume, which we will refer to as the "secondary plume", is located closer to the north-eastern margin of the glacier (Fig 1).

In what follows, we use bathymetry data from both years, while the other in situ observations were mostly collected during the 2013 season (see Stevens et al., 2016; Mankoff et al., 2016, for further details on the field campaigns).

### 2.1    Bathymetry

The bathymetry of Sarqardleq Fjord was first mapped in detail during the 2012 and 2013 field seasons and the immediate bay in front of the terminus was found to feature depths of 40 – 150 m (Stevens et al., 2016). These initial results were limited to data from a Remote Environmental Monitoring UnitS (REMUS) Acoustic Doppler current profiler (ADCP) and a shipboard ADCP, which did not get closer than ~ 200 m to the glacier front. Here, we supplement these data with several additional near-terminus

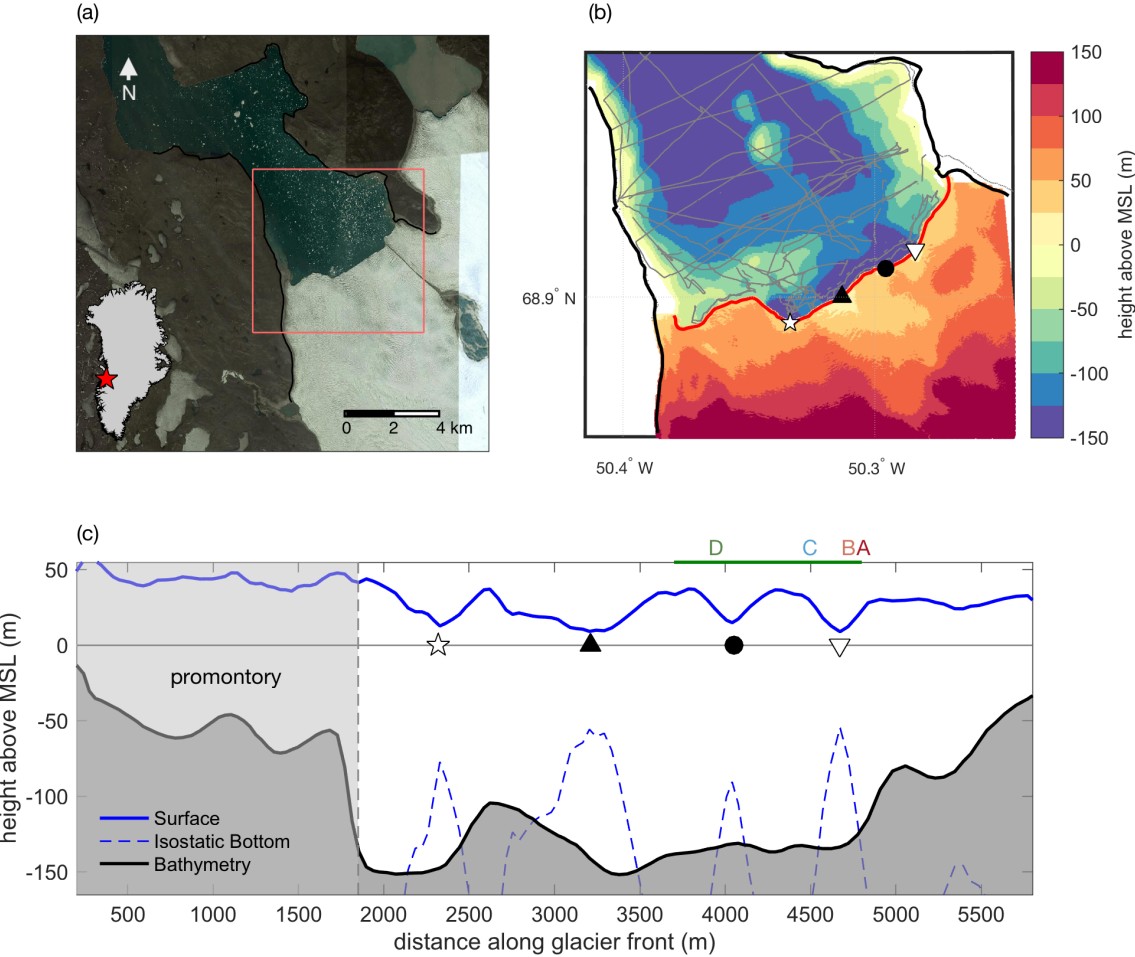

**Figure 1.** (a) Landsat-8 image of the lower part of Saqqarliup Sermia, and Sarqardleq Fjord. The inset of Greenland shows the location of the glacier. (b) Gridded bathymetry from in situ observations (readings indicated by gray dots). Also shown is the surface height from ArcticDEM (Digital Elevation Map created by the Polar Geospatial Center from DigitalGlobe, Inc. imagery). Red line shows the front position on 9 July 2013. (c) Surface height (blue) and bathymetry (black) along glacier front (following the red line in panel b). Also shown is the isostatic bottom of the ice (blue dashed). Locations of two main plumes are highlighted in panels (b) and (c) by ☆ and ▽; two additional surface dips are indicated by ▲ and ●. The green horizontal line above panel (c) and the letters A-D indicate the locations of the front profiles shown in Fig 8.

datasets from the 2013 field campaign (Fig S1), which allows for a detailed bathymetry map along the grounding line. The new data consist of circa 39,000 depth readings taken with Jetyak-mounted (Kimball et al., 2014) and ship-mounted ADCPs. In addition, there are approximately 6000 readings from a ship-mounted National Marine Electronics Association (NMEA)

bottom-range profiler and 6 readings from Expendable CTD sensors (XCTDs) deployed in the otherwise undersampled region of the main plume. Most of these readings are within 10–100 m of the glacier front.

Fig 1c shows the new bathymetry at the glacier front as a function of $x$, the distance along the glacier front. The bathymetry can be split into two main regimes: For $x < 1800$ m (the promontory) the glacier is grounded in shallow waters and its surface heights are elevated substantially above flotation. From here on, we refer to the eastern part of the glacier ($x > 1800$ m) as the "main" glacier. In 2013, the front of the promontory was grounded on a sill that runs parallel to the glacier front. This sill coincides approximately with the furthest advance of the glacier in 1992 (Stevens et al., 2016). By 2013 the main glacier had retreated ~ 500 m from the sill, but the promontory was still perched on it in bathymetry of 60 m depth or less (Fig 1c). Since 2013, this part of the glacier front has also retreated by several hundred meters (Fig S2). In 2013, the main part of the glacier front was in waters of depth up to 150 m. A pronounced dip in bathymetry – suggestive of a subglacial channel – is found near the location of the main plume ($x = 2000 - 2400$ m). A number of smaller dips are observed between $x = 3400 - 4700$ m. Beyond 4700 m the water depth decreases rapidly as one approaches the northeastern shoreline.

## 2.2  Glacier surface topography

We obtained a digital elevation map (DEM) from an ArcticDEM overflight on 22 March 2013, which covers the full span of the Saqqarliup glacier front and some of the upstream region (Fig 1). The DEM has a horizontal resolution of 2 m and is capable of resolving individual crevasses on the glacier surface.

The DEM shows that the front of the glacier is heavily crevassed and has several pronounced dips in the surface elevation at the terminus. The ice cliff is highest (up to 50 m) and most uniform in the region of the promontory, while the main part of the glacier is much more variable with four distinct depressions that reach below 10 m surface elevation (indicated by symbols in Fig 1).

The coincident high-resolution surface elevation and bathymetry data near the terminus enable us to compute the total ice thickness along the glacier front which allows for an estimation of the total ice flux (discussed in Section 3.1).

## 3  Components of the frontal mass balance

In order for the mass budget along the glacier front to be balanced, the sum of advective ice flux and frontal retreat must be balanced by total ablation (i.e., by the sum of melting and calving fluxes). Here we consider a steady state, vertically averaged balance. At a given point $x$ along the glacier front this can be written as

$$H\left(R + v_i\right) = D\bar{M} + C. \tag{1}$$

The left hand side represents retreat and advection, where $H$ is the ice thickness (in m), $R$ is the retreat rate, and $v_i$ the ice velocity at the terminus (both in m yr$^{-1}$). The first term on the right represents the ice loss due to submarine melting, where $D$ is the draft of the glacier (in m) and $\bar{M}$ is the depth-averaged melt rate (in m yr$^{-1}$). The final term, $C$, is the ice loss due to calving (in m$^2$ yr$^{-1}$). In this section we discuss the data used and assumptions made to estimate each term in detail.

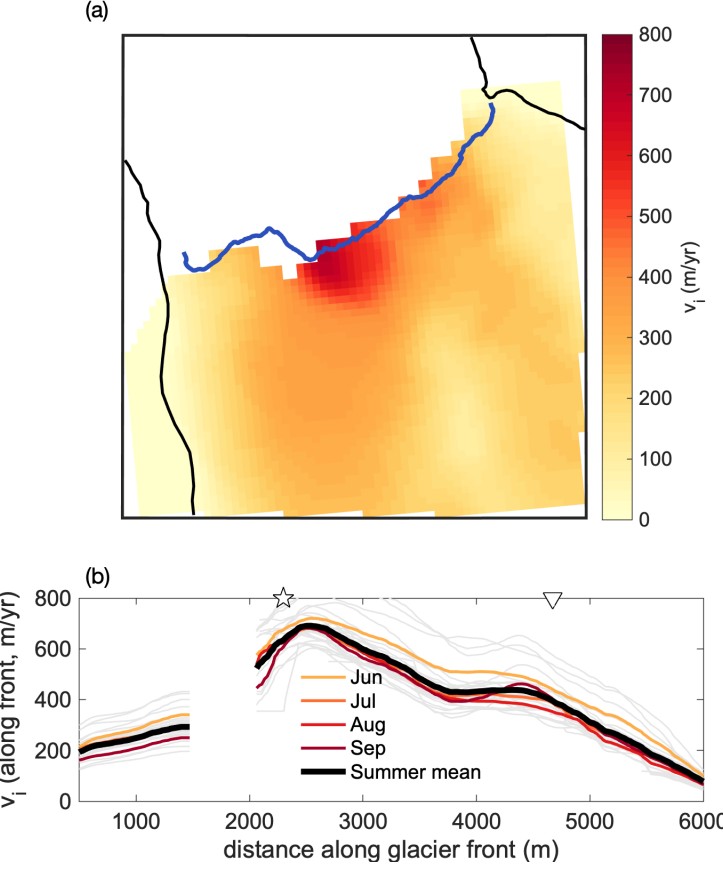

**Figure 2.** (a) InSAR ice velocity data near the glacier front. Shown are mean summer (June–September) values averaged over 28 velocity fields, collected during 2012–2014. Note that there is a consistent data gap near the promontory. The shading represents the horizontal velocity magnitude. (b) Velocity profiles along the glacier front. Here, as in all figures, the orientation is looking down-glacier. The faint gray lines show the 28 individual velocity fields. Also indicated are the approximate locations of the two plumes ($\star$, $\triangledown$).

### 3.1 Ice velocity and advective ice flux

Several dozen ice-velocity reconstructions of the lower part of the glacier are available for the years 2009 – 2015 from InSAR data (Joughin et al., 2011). The mean flow velocity at the glacier front (averaged over all available fields) is ~ 350 m yr$^{-1}$ with minima at the edges of the glacier. There is a notable peak in ice velocity (up to 750 m yr$^{-1}$) near the location of the main plume (Fig 2). A second region of elevated velocities is found near $x = 4500$ m and is more pronounced further upstream from the glacier front. The drainage location of this second ice stream coincides with that of the secondary plume. It is worth noting that the spatial distribution of velocities was remarkably consistent during summer months (June–September) from 2012–2014 (Fig 2b), followed by a substantial overall slowdown in 2015. This slowdown is not included here as it has been linked to a major drainage event of lake Tininnilik (Kjeldsen et al., 2017), and therefore is subject to altogether different environmental

forcing. In what follows, we will consider the 2012–2014 mean July velocity profile along the glacier front. Using the mean summer (June–September) velocities instead does not change the results appreciably.

The magnitude of the summer ice velocity along the glacier front, $v_i(x)$, shown in Fig 2b, together with the ice thickness profile $H(x)$, allows for an estimate of total advective ice flux (Fig 3). This assumes plug flow, i.e., that the ice velocity is approximately constant from the surface to the ice–bedrock interface. Note that for a glacier with no sliding and uniform temperature, the depth-averaged velocity is 80% of the surface velocity. For fast-flowing tidewater glaciers with concentrated deformation at depth, such as Saqqarliup, plug flow is therefore considered a good approximation (Meier and Post, 1987).

We note that the thickness data suggest that the terminus might be floating at several locations: the four highlighted surface depressions at the glacier front are all low enough to raise the isostatic bottom of the ice above the local sea floor. The locally-isostatic bottom of the ice is indicated in Fig 1c (blue dashed line). Here we assume an average ice density of 883 kg m$^{-3}$, obtained as a mean of low and high values commonly used for glacier and ice shelf front densities, namely 850 kg m$^{-3}$ (Silva et al., 2006) and 917 kg m$^{-3}$ (pure ice). The surrounding ice and the associated stiffness of the glacier will likely prevent the ice from assuming local isostasy along the glacier front. However, the isostatic bottom can be used to compute a lower bound on the ice thickness in regions where the ice may be floating. It may be speculated that the ice appears to be floating in these regions due to undercutting by submarine melt (which in turn is associated with rising discharge plumes, as discussed in Section 3.3). The ice would be grounded everywhere else. In particular, the ice surface is elevated substantially beyond its isostatic height in the region of the promontory. The uncertainty in ice thickness associated with the glacier potentially floating at several points is illustrated by the shaded areas in Fig 3. In the figure, the upper bound of the ice thickness assumes a fully grounded glacier front, while the lower (dashed) bound assumes local isostasy everywhere. The ice flux is highest when assuming a fully grounded glacier, while a partially floating glacier front would have a correspondingly reduced flux.

## 3.2 Changes in glacier front position

Superimposed on the aforementioned long-term retreat of the glacier front over the past decades (Fig S2) we observe a seasonal advance–retreat cycle during 2012 and 2013 (Fig 4). A total of 27 front positions between January–October 2012 and January–October 2013 were digitized from TerraSAR-X satellite images. The 15 profiles from 2013 are shown in Fig 4a. Both years exhibit a clear, albeit modest, seasonal cycle in terminus position, with a mean advance for the entire front of roughly 30 m from January through April/May, followed by a more rapid retreat from June to September of circa 80 m (Fig 4b). However, there is substantial variability along the glacier front in this cycle. Near the edges of the glacier, and in particular at the promontory, the glacier exhibits a much reduced advance–retreat cycle, and more variable regions are found in the main dynamic section of the glacier.

$R(x)$ is computed as the rate of retreat perpendicular to the initial glacier front. The most rapid retreat in 2013 was observed at the time of the July study period. Fig 4b shows the spatial-mean seasonal retreat anomalies for 2012 and 2013, with profiles from 9 July and 31 July 2013 highlighted in green. Such rapid retreat is spatially highly variable (Fig 4c) and strongly impacted by sporadic but large individual calving events. Longer-term mean retreat rates, computed from average spring and fall glacier front positions (highlighted in Fig 4 in red and blue, respectively) may therefore be more representative on longer time scales.

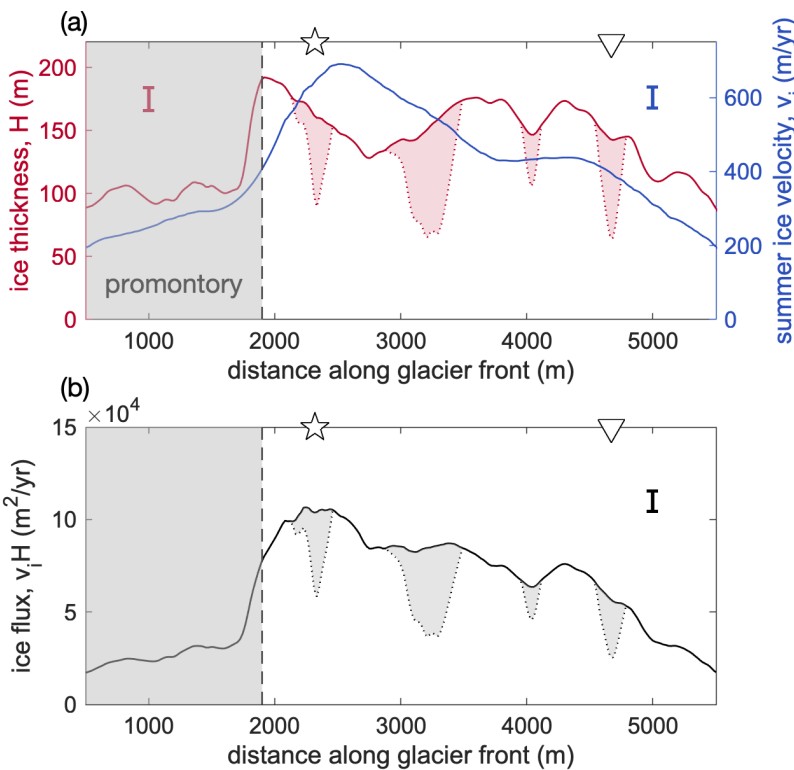

**Figure 3.** (a) Mean July ice velocity along the glacier front in blue (right vertical axis). Here we used cubic interpolation to fill the data gap shown in Fig 2b. In red (left vertical axis) is shown the estimated ice thickness along the glacier front, obtained by computing the difference of the surface and bathymetry profiles of Fig 1c. The dotted red line shows the ice thickness at the glacier front assuming the ice is locally in isostatic equilibrium everywhere. (b) Ice flux per unit width along the glacier front (in black), computed from the product of velocity and thickness (shown in panel a). The shaded gray areas under the curve show the ice-flux range due to potential flotation. This is a result of the thickness ranges indicated as red shaded areas in panel a. Uncertainties for thickness, velocity, and ice flux are shown by the red, blue, and black standard error bars, respectively. Also indicated are the approximate locations of the two known plumes ($\star$, $\triangledown$), which coincide with two areas of possible flotation.

### 3.3 Submarine melting

Submarine melt rates at Saqqarliup Sermia during summer 2013 have been estimated by Slater et al. (2018). Here we provide only a brief overview of the approach and build on the results of Slater et al. (2018) to investigate the glacier's flux balance. Melt rates within the two plumes were estimated using standard buoyant plume theory (Jenkins, 2011; Carroll et al., 2016; Slater et al., 2016). Melt rates outside of the plumes were estimated using a high-resolution numerical model of the fjord in the Massachusetts Institute of Technology general circulation model (MITgcm), which has become the leading model for simulating the circulation and water properties of glacial fjords and for estimating the resulting submarine melt rates (e.g. Xu

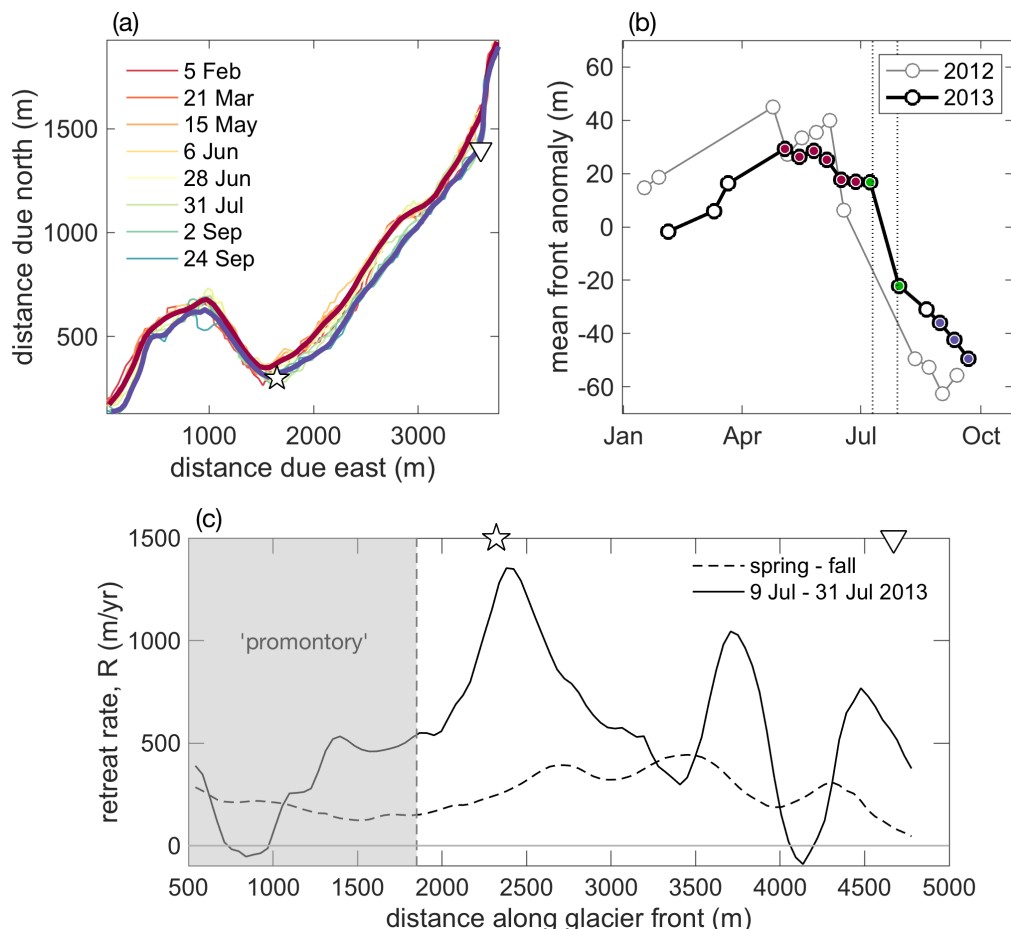

**Figure 4.** Seasonal advance and retreat of glacier front. (a) 15 front profiles acquired from February to September 2013; the legend lists every 2nd profile. The thick red and blue profiles represent the May–June and September averages, respectively. Also indicated are the location of the two plumes (☆, ▽). (b) Mean front position, shown as an anomaly from the yearly mean position. 2012 values are shown in gray, 2013 in black. The spring profiles used in panel (a) are highlighted in red, fall profiles in blue, July profiles in green. The vertical dotted lines demarcate the period from 12 to 31 July during which calving was observed. (c) Retreat rates, $R(x)$, along the glacier front. Positive $R$ represents glacier retreat and negative $R$ glacier advance. The dashed line represents the spring–fall 2013 mean retreat rates; the solid line that of 9 to 31 July, computed from the profiles marked green in panel b.

et al., 2012; Sciascia et al., 2013; Cowton et al., 2015; Carroll et al., 2015). Both buoyant plume theory and MITgcm were forced with runoff from the regional climate model RACMO and initialized with hydrographic profiles from the fjord. Slater et al. (2018) also presented observationally-inferred melt rates using water property and velocity measurements collected within 100 m of the calving front. In each approach, Slater et al. (2018) then used the standard three-equation melt rate parameterization of Holland and Jenkins (1999) to convert the modeled or observed water properties and velocities to an estimated submarine

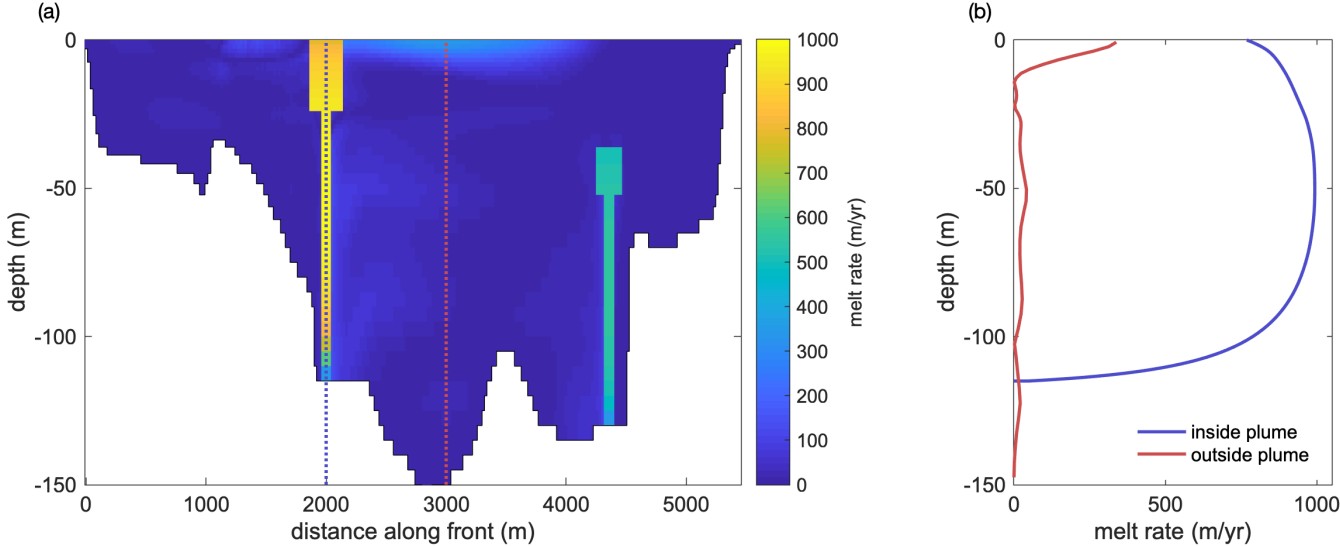

**Figure 5.** (a) Time-mean melt rates along the glacier front as estimated from MITgcm, adapted from Slater et al. (2018), their Fig 3f. The bathymetry in the model (white) is based on that of Stevens et al. (2016). (b) Melt rates averaged inside the main discharge plume (blue) and outside of both plumes (red).

melt rate. There is good agreement between the melt rates estimated with MITgcm and with observations (Slater et al., 2018, their Fig 3). Here, we only consider the modeled melt rates (Fig 5), which have the advantage of covering the whole extent of the glacier front (unlike rates inferred from observations, which have data gaps in and around the plumes).

There is large spatial variability in submarine melt rates along the glacier front (Fig 5). Submarine melt rates are highest
5    (both in a depth-averaged and maximum sense) within the two plumes where the discharge of buoyant surface meltwater from beneath the glacier gives high water velocities. Outside of the two plumes melt rates are much smaller in a depth-averaged sense, however the lateral circulation excited by the plumes combines with warm surface waters to give high melt rates near the surface outside of the plumes (Fig 5c; see also Slater et al., 2018).

While these melt rate estimates represent the state-of-the-art in terms of melt rate modeling, we stress that they are based on
10    a melt rate parameterization that has not been confirmed by observations, especially for the case of a mostly vertical front of a tidewater glacier. The uncertainty associated with these melt rate estimates is further discussed in Section 5.

### 3.4 Calving frequency and distribution

Calving events were detected over a 19-day period from 12 July to 31 July 2013, using two pressure sensor moorings located on the western and eastern banks of the fjord, each at a distance roughly 2 km from the nearest point along the glacier front
15    (Fig 6a). The dispersion of waves that are created by individual calving events can be inverted to estimate the distance between the mooring and the origin of the wave. Wave packets that are detected by both moorings can be used to triangulate the time

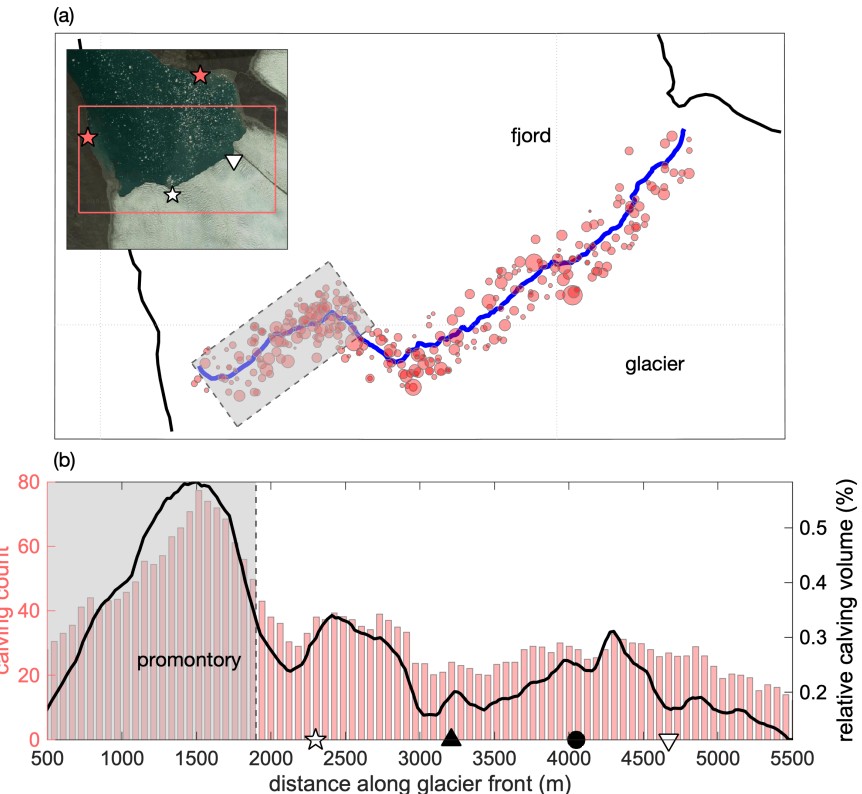

**Figure 6.** (a) Spatial calving distribution as estimated from pressure sensor data; the shaded rectangle indicates the promontory. Inset: Close-up of the glacier front and adjacent fjord, with the red rectangle outlining the region of interest and red stars indicating the location of the wave moorings (b) calving count along the glacier front, obtained as the total number of calving events detected within a 300 m running window along the glacier front (red bars, left axis). Also shown is an estimate for the relative calving volume, computed from the product of the frequency of calving events and the corresponding magnitudes of the detected waves (black line, right axes). Plume and surface dip locations are indicated as in previous figures.

and position of the corresponding calving event (Minowa et al., 2018). For the present dataset, this method has been validated against a photography-derived calving record and good correspondence was observed (not shown). The study by Minowa et al. (2018) provides a detailed description of the method.

In total, 336 calving events were identified using this method over the period that both sensors were recording. Fig 6a shows the location and wave amplitude of the individual events. The calving frequency distribution along the glacier front is illustrated in Fig 6b.

A pronounced peak in frequency is found at the promontory, where shallow bathymetry causes the glacier to be elevated substantially beyond its isostatic height of flotation. With its high ice cliffs the promontory can be regarded as a region that is subject to a rather different calving regime than the rest of the glacier.

For the main part of the glacier, we observe a peak in calving activity at a distance $x \approx 2400$ m along the glacier front, near the concave bend in the glacier front. A second peak in calving activity is found around $x \approx 4300$m. Both peaks appear slightly offset from the location of the two plumes. The calving activity is lowest at the northeastern edge of the terminus.

Even though this dataset presents a rather accurate record of calving frequencies, it remains challenging to infer a total volume of calved ice (Minowa et al., 2018). This is due to the different modes of calving (e.g., ice-cliff calving versus submarine calving), as well as the different shapes of calved ice blocks and the differing heights from which they fall (or depths from which they rise). Distinguishing between these events from the pressure sensor data is a difficult task and beyond the scope of this study. The pressure sensors do record an amplitude of the incoming wave packet associated with a given calving event. Crudely approximating that this amplitude is proportional to the size of the calved ice we can estimate a relative calving volume (black curve in Fig 6b). However, since a small cone-shaped ice block can act as a more efficient wave generator than a large flat piece of ice (N. Pizzo, personal communication, Bühler, 2007), it is difficult to ascertain a direct relation between wave amplitudes and calving volume. In what follows we therefore only consider the calving frequency record and will scale this record such that the resulting calving flux approximately closes the mass budget at the glacier front (see Section 4.2). Given the limitations of the data, we take such a scaling to be the most justifiable first-order approximation, supported by the rather uniform distribution of estimated calving event sizes along the glacier front. The scaling factor is chosen such that the mean calving volume is equal to the mean of the residual, i.e. $\langle C \rangle = \langle H(R + v_i) - D\bar{M} \rangle$, where $\langle \rangle$ denotes the spatial mean along the glacier front.

## 4 Overall flux balance and spatial variability

In what follows we consider the volume flux across the glacier front during the summer of 2013. We make the assumption that this flux was steady during the study period and ignore time-dependencies of the individual terms in equation (1).

To compare the different terms in the mass budget, we consider the retreat rate as computed from the two fronts measured on 9 and 31 July 2013, since this is almost the exact time window of the calving observations (12 – 31 July). For the advection term we use the July average over the years 2012 – 2014, since the July 2013 ice velocity fields have substantial data gaps at the glacier front. However, as discussed above, there is little interannual variability in $v_i$ over these years, so the 3-year mean likely gives a close approximation to the July 2013 velocity field. Front retreat and advective flux along the glacier front (i.e., the left hand side of equation 1) are shown in Fig 7a. The sum of ice advection and retreat is compared to the estimated melt fluxes in Fig 7b. Fig 7c shows the calving flux as estimated from the observations (Section 3.4), compared to the residual $C$ of the other three terms in equation 1, such that $C = H(R + v_i) - D\bar{M}$.

### 4.1 High spatial variability along the glacier front

A striking feature of almost all components of this multipartite dataset is their high spatial variability along the glacier front.

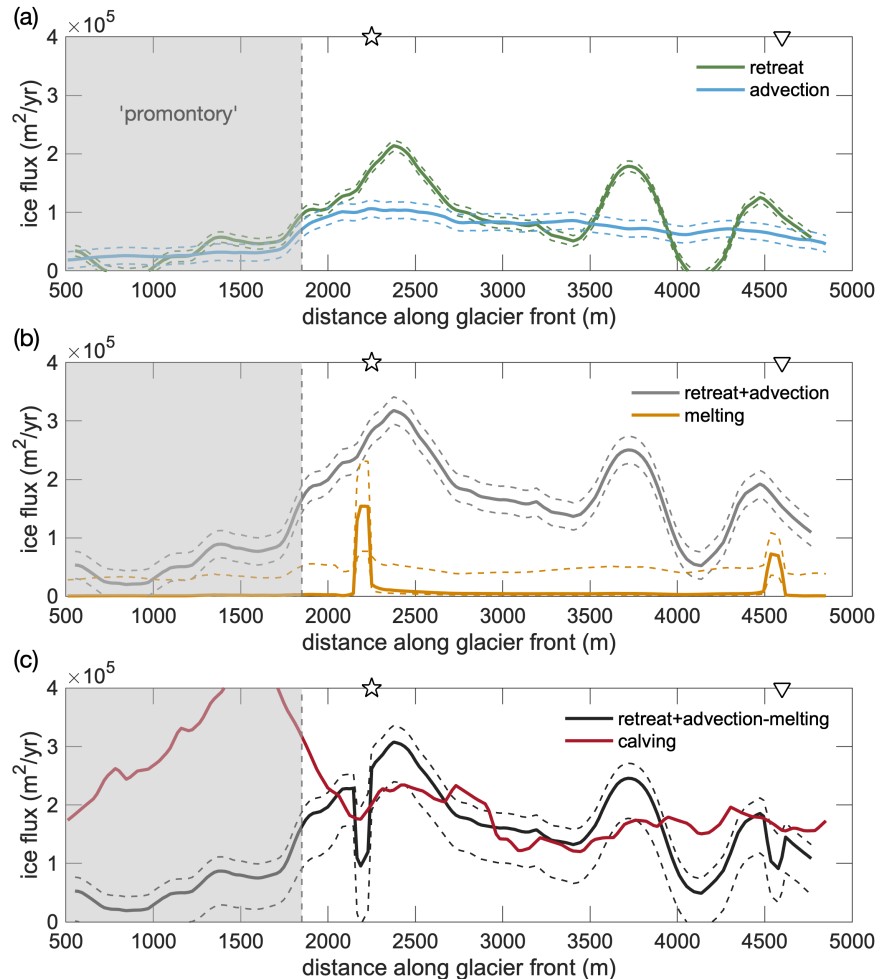

**Figure 7.** Flux balance along the glacier front. Dashed lines indicate uncertainties as discussed in the text. (a) The green line represents the July 2013 retreat rate and the blue line the advective ice flux. (b) Sum of retreat and advection (gray) and melt flux (orange). (c) Approximate closure of the volume flux budget along the glacier front. The black line shows the residual of advection plus retreat minus melting, while the red line shows the observational calving estimate as in panel b. Note that the calving flux has been scaled to approximately close the budget for the main part of the glacier.

Away from the margins, the ice thickness at the front ranges from thin (<40 m) sections near the northeast edge to ~100 m along the promontory and up to 192 m near the main plume, with substantial variations throughout. Overall, we observe a mean thickness of 128 m with a variability of ±38 m (one standard deviation).

We find that the advective flux is the most uniform component, still it is notably suppressed at the promontory and highest near the outflow location of the main plume (Figs 3, 7a).

The retreat rates are overall of comparable magnitude to the advective flux. However, the retreat rates are spatially extremely variable, in particular the observed July 2013 rates, which exhibit three regions of enhanced retreat, two of which are close to the two discharge plumes, with peaks at $x = 2400$ and $4400$ m (Figs 4c, 7a). Averaged over longer time periods, the retreat rates become more uniform. Over shorter time periods retreat rates are more strongly influenced by individual calving events.

The melting estimates feature two pronounced maxima at the plumes and are small everywhere else (Fig 7b). The maximum melt flux value at the main plume ($1.5 \times 10^5$ m$^2$ yr$^{-1}$) is slightly higher than the mean retreat and advective flux values ($1.0 \times 10^5$ and $0.8 \times 10^5$ m$^2$ yr$^{-1}$, respectively). Outside the two plumes the mean melting flux ($0.04 \times 10^5$ m$^2$ yr$^{-1}$) is an order of magnitude lower than inside the plume and than the other budget terms. Dividing these depth-integrated flux values by the average thickness (128 m), we obtain depth-averaged velocities for each term. These are ice advection: 780 m yr$^{-1}$; retreat:

620 m yr$^{-1}$; maximum melt rate at main plume: 1200 m yr$^{-1}$; and mean melt rate outside the plumes: 30 m yr$^{-1}$.

Calving frequencies are strongly enhanced at the promontory, which – given the reduced advection and retreat in this area – implies that calved pieces are in general smaller here. Since we are unable to adequately distinguish between the different calving sizes, the heightened calving activity at the promontory results in a large discrepancy between the computed residual $C$ and the observationally estimated calving flux in that region (Fig 7c). We may also be underestimating the advective flux

at the promontory slightly, since we only consider horizontal velocities, and the ice flow may have a non-negligible vertical component as the glacier rides onto the local sill. Even though calving frequencies are overall lower for the main part of the glacier, we observe two local maxima, slightly offset from the plumes (Fig 7b). The lowest calving frequencies are found between the two plumes in the region farthest from both plumes. The two peaks in depth-averaged melt flux (Fig 7b), co-located with the two discharge plumes, are just offset from the two peaks in frontal retreat and calving.

**4.2 Spatially integrated mass budget**

Integrated along the main part of the glacier front we estimate an ice advection rate of $0.2 \pm 0.05$ Gt yr$^{-1}$ and a retreat rate of $0.3 \pm 0.03$ Gt yr$^{-1}$. This gives as a best estimate for the total rate of ice loss $\sim 0.5$ Gt yr$^{-1}$. The uncertainty in ice advection corresponds to one standard deviation in the spread of mean July ice velocities. The uncertainty in the retreat rate is largely due to the somewhat arbitrary selection of "before" and "after" dates, and the resultant disproportionate impact of individual

calving events. The error reported here is one standard deviation in the difference in retreat when choosing the frontal profiles of June 28 (instead of July 9) as "before" date or August 22 (instead of July 31) as the "after" date.

Integrating the estimated melt over the main glacier front gives a total melting flux of $0.03$ Gt yr$^{-1}$. This would suggest that $\sim 0.47$ Gt yr$^{-1}$ (or $94\%$) of ablation occurs in the form of calving, thus implying that the glacier balances the ice flux almost exclusively through calving (except in the narrow regions at the discharge plumes). The lack of an ice mélange in the fjord, and

the anecdotal observation of limited calving are, however, at odds with this finding. This raises the question whether the melt term – estimated using state-of-the-art parameterizations informed by observations very close to the ice front – is incorrect? This is discussed further in Section 5.

While we have no direct measurement of calving volume, we can close the integrated mass budget by scaling the observed relative calving frequencies to give the required total calving flux of $0.47$ Gt yr$^{-1}$ (Fig 7c). This corresponds to a mean calving

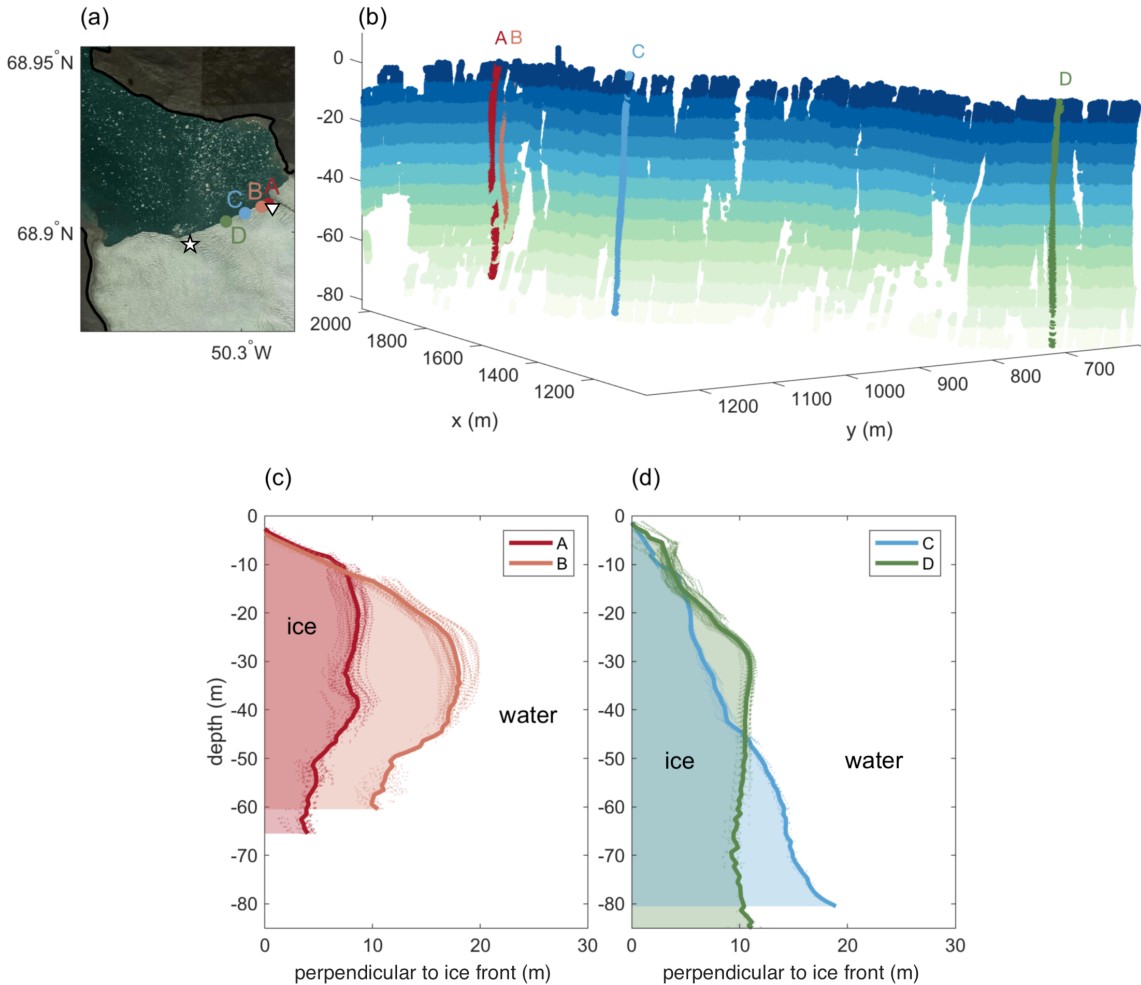

**Figure 8.** Multibeam sonar data of glacier front from 26 July 2013. (a) Map illustrating the location of the multibeam cross-sections A–D and the two plumes ($\star$, $\triangledown$). (b) 3D point-cloud transect showing a part of the eastern side of the glacier (distance along glacier front, ~ 4000 − 4800 m). Data is color-coded by depth below sea level. Indicated are the locations of the four cross-sections A–D shown in panels c and d. (c) Cross-sections A and B near subglacial plume, exhibiting characteristic undercutting. (d) Cross-sections C and D away from plume, showing submarine protrusions without undercutting.

flux of $1.7 \times 10^5$ m$^2$ yr$^{-1}$ along the glacier front (compared to a mean melting flux of $0.1 \times 10^5$ m$^2$ yr$^{-1}$). Again dividing by the average thickness, this corresponds to 1300 m yr$^{-1}$ ice loss due to calving, compared 80 m yr$^{-1}$ of melting.

## 4.3 Variations in the vertical glacier front profile

A final piece of observational evidence which may help in the interpretation of the results above is provided by point cloud images of the glacier front profile. These were collected during the 2013 field season using an autonomous surface vehicle,

the Woods Hole Oceanographic Insitution "Jetyak" (Kimball et al., 2014; Mankoff et al., 2016). Among other instruments, the Jetyak carried a multibeam sonar that was mounted sideways facing the glacier, which collected three-dimensional maps of the underwater portion of the glacier front. Further details of the Jetyak's operation and data can be found in Kimball et al. (2014). Here, we highlight several characteristic frontal profiles. Fig 8 shows a point-cloud transect of the northeastern flank of the glacier, as well as four vertical line profiles at different locations along the transect.

The first two profiles (A and B) are placed near the secondary plume. Both profiles are marked by two features: (i) a sloped upper 20–25 m, which results in the above-water cliff of the glacier being set back by 10–20 m, relative to the most ocean-ward point of the glacier face. (ii) Below 40 m depth we observe up to 10 m of undercutting, such that the protrusion beyond the above-water cliff is most pronounced at depths 20–40 m, and the ice is substantially eroded at greater depths. This is likely caused by the rising subglacial plume which leads to preferential melt of the deeper parts of the glacier front (Fried et al., 2015; Slater et al., 2017). Note that the high turbidity of water within the main plume prevented the Jetyak from surveying the shape of the glacier front occupied by that plume.

Profiles C and D, which are located far from the plume, also feature said underwater ice protrusion, however, they show no signs of undercutting. The presence of such net-buoyant underwater protrusions and their potential impact on calving has been studied previously (Wagner et al., 2014, 2016; Sugiyama et al., 2019), and will be discussed further in the next section.

We note that the bathymetry reaches depths of around 130 m for this part of the glacier and the bottom ~ 50 m are unfortunately not captured by the multibeam sonar. However, the profiles located near the melt (A and B) can be expected to be further undercut below the observed range (Fried et al., 2015), while profiles C and D likely do not feature such undercutting.

## 5   The role of melting in the frontal mass budget

### 5.1   Uncertainty in melt rate estimates

The finding that calving appears to make up almost the entire loss of ice is somewhat unexpected, in particular since during the study period the glacier's calving activity was limited to relatively small events, and the fjord was by-and-large devoid of icebergs. Furthermore, the melt rates used here are roughly double that of what previous estimates would have been since we account for additional melt that arises from the recirculation of warm ambient surface waters (Slater et al., 2018). Yet, melting supposedly only makes up ~ 6% of the total ablation.

Given the lack of observational verification of the current melt rate parameterization, it is worth considering end-member melt rate scenarios. A key parameter in the melt rate parameterization is the thermal Stanton number, which directly controls the rate of transfer of heat from the ocean to the ice. Its canonical value is based largely on field observations at a cold Antarctic ice shelf and there are as yet no strong observational constraints from tidewater glaciers. Furthermore, Ezhova et al. (2018) have recently argued for a larger Stanton number based on direct numerical simulations. We thus consider lower and upper bounds for melt rates in which the thermal Stanton number is respectively reduced and increased by 50% (Fig 7b,c).

To obtain the upper bound melt rate scenario, we also increase the outside-of-plume water velocity which enters the melt rate parameterization. While vertical velocities inside of plumes might be considered reliable based on well-validated plume

theory (Morton et al., 1956), one could argue that the mean modeled outside-of-plume velocities may be too small for a number of reasons including coarse model resolution and the lack of tides, surface waves, and calving events which may excite water motion. These factors might crudely be taken into account by placing an additional velocity in the melt rate parameterization. Such an approach has some precedent with the inclusion of tides beneath ice shelves (Jenkins and Nicholls, 2010). In the upper

bound melt rate scenario, we thus add 0.2 m s$^{-1}$ to the outside-of-plume water velocity entering the melt rate parameterization.

In the lower bound melt rate scenario, melting accounts for an even smaller fraction of mass loss than in our best estimate, but is still significant inside the plumes. In the upper bound melt rate estimate, melting accounts for a significant proportion of mass loss both inside and outside of the plumes (Fig 7b). Clearly this is an observationally under-constrained discussion, and we emphasize that these upper and lower bounds are very rough error estimates as the state of understanding of submarine

melting does not yet permit rigorous quantitative assessment of uncertainties. Yet these bounds do show that through reasonable modification of the melt rate parameterization, melting can account for a larger fraction of the ice loss than reported in our best estimate. Even by introducing these uncertainties, however, the analysis presented still indicates that calving is the dominant mode of mass loss for most of the glacier front (except at the localized melt plumes).

## 5.2  Impact of melting on calving

In addition to balancing the frontal ice flux, the data allow us to examine how melting and calving may be interlinked. Specifically, one consequence of melting being focused on narrow regions is that it can lead to sharp incisions in the glacier front, which in turn may enhance calving.

Slater et al. (2018) found that fjord-scale circulations driven by plumes can result in enhanced submarine melting near the fjord surface in regions distant from the plume Fig 5b). This near-surface melting has in turn been suggested as a potential

driver for large calving events at glacier fronts that are floating or close to floating (Wagner et al., 2016): Preferential near-surface melting at the glacier front leads to a horizontal melt incision near the water surface which in turn causes erosion of the above-water ice cliff. As a result, the front of the glacier is left with an underwater protrusion (or "ice foot") as in the profiles C and D of Fig 8. This frontal profile is statically unstable, since the ice foot is net buoyant and exerts bending stresses on the glacier (Robertson et al., 2012; Benn et al., 2017). Calving events occur when such stresses surpass the yield strength of the

terminus. It is likely that profiles C and D represent sizable ice feet which exert bending stresses that enhance the calving flux in this region.

Furthermore, it is possible that the regions adjacent to the meltwater plumes are more prone to calving since the high melt rates at the plumes cause vertical incisions in the glacier front (Fried et al., 2015). These in turn would reduce the transverse (i.e., along-front) stability of the terminus, and trigger further calving. A surface expression of such a vertical incision in the

glacier front can be found near the main plume in the profile of August 2012 (Fig S2). Considering the particular geometry of Saqqarliup, as the two main plumes drive rapid melt near the two edges of the main part of the glacier, this may cause the entire front between the plumes to be more prone to calving, in particular since we have found this region to be close to (or at) flotation.

In summary, from the observations presented in the previous sections, we propose that there are two distinct regimes driving ablation at Saqqarliup: (a) melting-dominated ablation in spatially confined regions near the discharge plumes, and (b) calving-dominated ablation in the regions away from the plumes (which may be enhanced by near-surface horizontal melt incisions). This is further supported by the local minima in calving activity at the location of the two discharge plumes (Fig 7b). The two ablation regimes are summarized in the schematic of Fig 9.

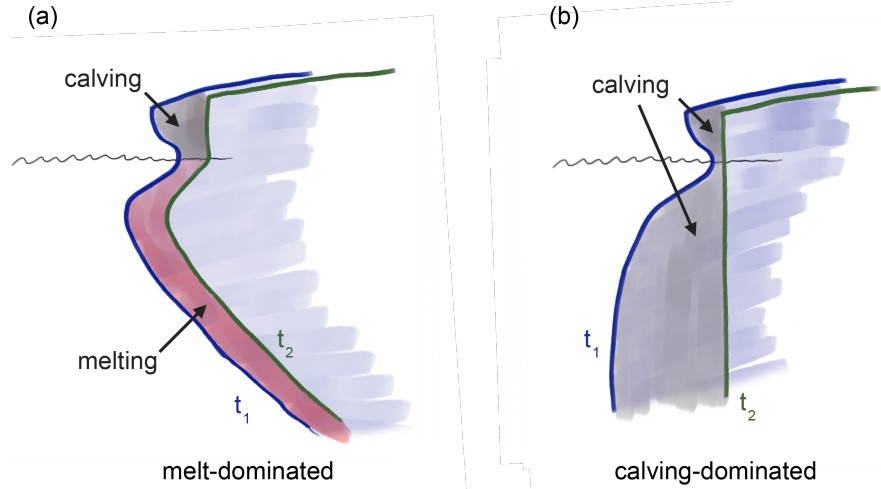

**Figure 9.** Schematic of two distinct ablation regimes. (a) Melt-dominated regime: the vertical structure of melting due to a rising subglacial discharge plume which entrains warm ambient water results in substantial undercutting of the glacier front (as in profiles A and B in Fig 8). These front profiles likely do not cause large calving events with calving mostly confined to the smaller above-water cliff. Profiles are drawn for an earlier time $t_1$ and a later time $t_2$ by which the glacier has retreated mostly due to melting. (b) Calving–dominated regime: here the growth of sizable and buoyant underwater feet (as in profiles C and D in Fig 8) can accelerate calving, with the melt contribution confined to a small region near the water surface. Again, profiles are shown at $t_1$ and $t_2$ (pre and post-calving), as part of the "footloose" calving cycle (Wagner et al., 2014).

## 6    Conclusions

We have presented a multi-faceted dataset of a Greenland tidewater glacier and its surroundings. The unique dataset enables us to investigate the individual terms that determine the flux balance along the glacier front.

We find that the individual terms that comprise the glacier's frontal mass budget are marked by high spatial variability. Ice velocities feature maxima that coincide with troughs in the bathymetry and locations of subglacial discharge plumes. The retreat rates are spatially particularly variable when calculated over shorter periods of time (days to weeks) and are likely dominated by somewhat stochastic calving events over such short timescales.

Estimated submarine melt rates from numerical modeling of fjord circulation show rapid melting within the two discharge plumes and more widely at the fjord surface, but limited melting elsewhere. If we use the inferred melt rate to scale the calving

flux we find that 94% of the mass balance of this glacier must be balanced by calving. This finding appears to be at odds with the observation of limited calving and the lack of icebergs in the fjord. We suggest that the numerical model – even though constrained by direct measurements and using the standard melt parameterization – may underestimate melting outside of the plumes, indicating that current melt models for tidewater glacier fronts may need to be reviewed and should be treated with caution.

The spatial variability of the observed processes suggests the presence of two distinct ablation regimes: a melting-dominated regime near the discharge plumes and a calving-dominated regime away from the plumes. We discuss that melting, through its horizontal and vertical variability, may play an important role in driving calving, thus having a dynamic effect out of proportion to the fraction of mass lost by melting. If calving is indeed dependent on the localized melt rates, this may have far-reaching implications for the overall stability of the glacier. Understanding the impact of these spatially highly variable processes on ice sheet dynamics should thus be a priority in the study of ice–ocean interactions.

*Competing interests.* The authors declare no competing financial interests.

*Acknowledgements.* We acknowledge support from the Woods Hole Oceanographic Institution Ocean and Climate Change Institute Arctic Research Initiative, and NSF OPP-1418256 and OPP-1743693, to F. Straneo and S. B. Das. T. J. W. Wagner was further supported by NSF OPP award 1744835. Geospatial support for this work provided by the Polar Geospatial Center under NSF OPP awards 1043681 and 1559691. DEMs provided by the Polar Geospatial Center under NSF OPP awards 1043681, 1559691 and 1542736. D. A. Slater acknowledges the support of Scottish Alliance for Geoscience, Environment and Society early-career research exchange funding.

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
