# Peer review of "Large spatial variations in the flux balance along the front of a Greenland tidewater glacier"

_The Cryosphere, 2018_

## Referee Comment (RC1) · Anonymous Referee #1 · 6 Sep 2018

Review of: Large spatial variations in the frontal mass budget of a Greenland tidewater glacier Authors: Wagner et al.

This study uses field data, satellite observations, and numerical modeling results to investigate spatial variations in iceberg calving and submarine melting. This knowledge is useful because it provides information on how submarine melting might affect calving and terminus retreat. I think this is an interesting way of thinking about the ice-ocean interface.

The paper essentially boils down to an analysis of a mass continuity equation (eq. 1 in the manuscript), in which the rate of terminus retreat is related to the glacier velocity, calving rate, and submarine melt rate.

The left-hand side of equation 1 contains the ice thickness, the rate of retreat, and the ice velocity. These quantities were determined using fairly traditional methods, and as such I feel confident in the results. I have much more trouble with the right-hand side of equation 1, which contains the submarine melt rate and calving rate. Specifically:

1. The submarine melt rate was determined by some combination of hydrographic observations and numerical modeling. How this was done is not clear, as the modeling is apparently presented in a separate paper that is in review. Without access to that paper and very little description of the model, I am basically asked to take the model results at face value. Even if the modeling paper was already published I would still appreciate to have more details of the model in this paper.

2. The calving rate was estimated by (i) locating calving events with measurements of ocean waves and (ii) converting measurements of calving events into calving rates by somehow scaling the number of events (and amplitude of the waves) so as to roughly balance equation 1. Given the wide variety of types of calving events and iceberg geometries and poor understanding of wave generation by calving events, it seems dangerous to assume any relation between wave amplitude and calving event size.

Using this method, the authors are unable to close the mass budget (see Fig. 6c) along the southern part of the terminus, which they attribute to the style of calving that they observe there. That's fine, I suppose, but elsewhere along the terminus the left- and right-hand sides of equation 1 differ by a factor of 2 or more, which raises questions about the validity of their results — and in particular the modeled melt rates and the estimated calving rates.

Its not clear to me why the calving rate wasn't just calculated by subtracting the melt rate from the left-hand side of equation 1, which would at least ensure that mass is being conserved. Nonetheless, a major conclusion in the paper is that frontal ablation is dominated by melting in a couple of specific locations but is otherwise dominated by calving. That is an interesting result, but I think that point can be made much more

simply, without the need to convert wave amplitudes to calving event sizes, and may already be apparent in the modeling paper that is in review. For example, in figure 6b it is apparent that the modeled melt fluxes are highly focused on certain regions of the terminus, which already implies that the calving fluxes must be comparatively large everywhere else.

I include some detailed comments below, focusing on areas where I found the descriptions to be somewhat ambiguous. In addition, all errors are discussed in the text, I think it would be helpful to include error bars or confidence intervals in the figures (especially figure 6).

Specific comments: p. 1, l. 14: Tidewater glaciers aren't boundaries per se but rather a part of the ice sheet. Also, tidewater glacier termini (and the bottom of the remaining ice shelves) are really the only boundary between the Greenland Ice Sheet and the ocean.

p. 2, l. 1-2: Its a little confusing how these sentences jump from ablation (in general) to specifically talking about frontal ablation.

p. 2, l. 15: You could also mention that ocean surface gravity waves (both short- and long-period) have been used to observe calving events in Greenland in previous studies.

p. 2, l. 26: It wasn't clear to me how the hydrographic data was used to infer melt / constrain the submarine melt model.

p. 4, l. 1: In what way is the plume amplified? The drainage event is relatively short-lived, so I suspect it wouldn't have an impact on the plume over time-scales longer than about a week.

p. 4, l. 28: Did you correct the DEM to account for the difference between the ellipsoid and the geoid?

Figure 2, caption: It doesn't really make sense to talk about the flux being into the page

here. Do you mean that the velocity profile is from the perspective of somebody looking down glacier?

p. 5, l. 11: Speculation shouldn't be part of the methods section.

Figure 3b (and elsewhere): I'm not sure if its correct to refer to this quantity as a flux. Should it be a flux per unit width?

p. 7, l. 7: You can be more specific here. For an infinite slap with no sliding and uniform temperature, the depth-averaged velocity is 80% of the surface velocity. The percentage goes up for rapidly sliding glaciers and those that have concentrated deformation at depth, such as tidewater glaciers.

Figure 4c: For clarity, considering specifying that negative retreat rates indicate advance. (My personal preference is to plot dL/dt, so that positive indicates advance and negative retreat, but this is fine too.)

p. 8, l. 9: Don't you mean across-glacier variability?

p. 9, l. 1-2: Do you actually calculate the retreat rate perpendicular to the initial terminus, or is it perpendicular to a straight line fit through the initial terminus?

p. 9, l. 8-14 (and elsewhere in the paper): "rates" and "fluxes" are conflated in several places, which may be confusing. For example, in equation (1), H*R is not the retreat rate.

p. 9, l. 25: derived calving record by... ? Should "by" be deleted? Or is something missing here?

p. 10, l. 16: How do you do this scaling? Through a minimization procedure?

p. 10, l. 13-15: Laboratory experiments by Burton et al. (2012) suggest that the wave amplitude of waves produced by capsizing icebergs depends on the energy released during the capsize event, which scales nonlinearly with iceberg geometry.

p. 10, l. 24: I assume that the modeled melt rates are constrained by hydrographic observations. Is that correct?

p. 12, l. 21: Doesn't the thickness have to go to 0 at the margins?

p. 12, l. 29-30: Seems self-evident to me.

p. 14, l. 4-5: Can you quantify the amount of ice in the fjord and how it varies seasonally (even in a rough sense by looking at satellite imagery)? This glacier is pretty slow, so the calving fluxes would have to be pretty small.

---

## Referee Comment (RC2) · Anonymous Referee #2 · 7 Sep 2018

Overall I think this is an interesting paper. It's a tricky topic, but an important one, and so it is nice to see people try to improve our knowledge. However, I do have a couple of major concerns about the fundamental approach. The authors are essentially trying to determine the balance between ice losses from calving and those from melting. However, it seems to be that both of these things are poorly constrained:

The melt rates are calculated using a model and observations, but these are from Slater et al., which is currently 'submitted'. As such, it is not accessible to me and I also think it is inappropriate to treat this as accepted, when it hasn't been peer-reviewed. I'm not saying this is the case, but if Slater et al were to be rejected or the technique deemed in appropriate, it would mean that this paper also had the same issues.

The other term is calving, for which the authors have frequency (which is a useful

dataset) but no volume / area data. They therefore take it as the residual of the left hand side of the equation, minus melting. As noted above, the melt calculations may have quite high errors and the method is actually not published. As such, I think this limits the confidence we can have in the results.

The approach of using waves to ID calving events has been demonstrated in Minowa et al 2018 but they also had time-lapse imagery, which meant they could determine the type of calving (topple etc) and they did not need size. Here, I think the size is needed, so that the calving term is more robust.

A more minor point is the paper structure: the methods and results are mixed together. I'm not totally against this and I can see why the authors have taken the approach here (as you need to get the results from one part to do the next), but I think it does reduce the accessibility for someone trying to repeat the experiments, who just wants to get at the methods. As noted above, I think it needs to at least briefly describe the methods in Slater et al submitted here (to avoid the reader having to go and find another publication). Slater et al should also be through the peer review process before it is cited, as it's central to the argument.

Finally, I think the abstract is a bit misleading over time frames. You have the calving data for July, but the earlier material implies that this is a year-round budget.

I've attached some minor comments in the annotated pdf.

Please also note the supplement to this comment:
https://www.the-cryosphere-discuss.net/tc-2018-143/tc-2018-143-RC2-supplement.pdf

**Supplement:**

[revised manuscript text omitted]

---

## Author Comment (AC1) · 21 Nov 2018

Please find attached: response to reviewers document, revised manuscript with tracked changes, revised manuscript without tracked changes, supplementary information file

Please also note the supplement to this comment:
https://www.the-cryosphere-discuss.net/tc-2018-143/tc-2018-143-AC1-supplement.zip

---

## Author Response (AR1)

**Response to Reviewers** (reviewers' comments in italics)

We would like to thank the reviewers for their constructive and detailed feedback. We have substantially revised the manuscript in response to the comments provided and believe that the current version is much improved.

Both reviewers commented (i) on our methodology to constrain the ice flux from calving, as well as (ii) on the lack of support for the melt estimates (since we largely relied on a then yet-to-be-published study). We have addressed these issues as follows:

i)      We provide more detail now on the assumptions that go into the calving estimates and highlight the associated uncertainties. Following the suggestions of the reviewers we now calculate the calving flux as a residual of the other terms in the flux balance and discuss the associated uncertainty by comparing it to the independent estimate obtained from the pressure sensor data. We further have added error bars to the plots.

ii)     The paper by Slater et al from which the melt estimated are derived has since been published at Geophysical Research Letters and can be accessed here: https://agupubs.onlinelibrary.wiley.com/doi/10.1029/2018GL080763. We have also included more detailed descriptions of the melt modeling efforts in the manuscript, to ensure that it is a stand-alone piece of work. We have further placed the study more carefully in context of the existing literature on melt modeling.

We have also restructured the paper somewhat to more closely reflect the standard sequence of intro, methods, results, discussion, conclusion, as recommended by the reviewers.

Below we respond to each of the individual comments by the reviewers.

**Reviewer #1**

*This study uses field data, satellite observations, and numerical modeling results to investigate spatial variations in iceberg calving and submarine melting. This knowledge is useful because it provides information on how submarine melting might affect calving and terminus retreat. I think this is an interesting way of thinking about the ice-ocean interface.*

*The paper essentially boils down to an analysis of a mass continuity equation (eq. 1 in the manuscript), in which the rate of terminus retreat is related to the glacier velocity, calving rate, and submarine melt rate.*

*The left-hand side of equation 1 contains the ice thickness, the rate of retreat, and the ice velocity. These quantities were determined using fairly traditional methods, and as such I feel confident in the results. I have much more trouble with the right-hand side of equation 1, which contains the submarine melt rate and calving rate.*

We agree with the reviewer that the right-hand side of eq 1 (i.e. the melt and calving terms) is less well constrained than the left-hand side. At the same time understanding the partitioning of melting at calving fronts is key to improving our ability to understand and predict future ice sheet change. Here, we use a unique data set and strive to estimate the relative contribution of melting and calving along the entire glacier for the first time. We are aware of the uncertainties and have made every effort to spell these out, especially in the revised manuscript. At the same time, we feel that the results presented – large spatial variability along the glacier and dominance of calving versus melting – are sound results that are supported by the evidence presented even when the uncertainties are taken into account.

*Specifically:*

*1. The submarine melt rate was determined by some combination of hydrographic observations and numerical modeling. How this was done is not clear, as the modeling is apparently presented in a separate paper that is in review. Without access to that paper and very little description of the model, I am basically asked to take the model results at face value. Even if the modeling paper was already published I would still appreciate to have more details of the model in this paper.*

We thank the reviewer for raising this point. As mentioned above, the paper by Slater et al has since been published and can be accessed here: https://agupubs.onlinelibrary.wiley.com/doi/10.1029/2018GL080763.
We have further added a more detailed discussion of the melt-rate estimation, as well as a melt-specific figure (Fig 5), which will allow for an independent assessment of the melt-rate modeling results. The present manuscript thus becomes a stand-alone piece of work. The added section on melt reads as follows:

"Submarine melt rates at Saqqarliup Sermia during summer 2013 have been estimated by Slater et al. (2018). Here we provide only a brief overview of the approach and build on the results of Slater et al. (2018) to investigate the glacier's flux balance. Melt rates within the two plumes were estimated using standard buoyant plume theory (Jenkins, 2011; Carroll et al., 2016; Slater et al., 2016). Melt rates outside of the plumes were estimated using a high-resolution numerical model of the fjord in the Massachusetts Institute of Technology general circulation model (MITgcm), which has become the leading model for simulating the circulation and water properties of glacial fjords and for estimating the resulting submarine melt rates (e.g. Xu et al., 2012; Sciascia et al., 2013; Cowton et al., 2015; Carroll et al., 2015). Both buoyant plume theory and MITgcm were forced with runoff from the regional climate model RACMO and initialized with hydrographic profiles from the fjord. Slater et al. (2018) also presented observationally-inferred melt rates using water property and velocity measurements collected within 100 m of the calving front. In each approach, Slater et al. (2018) then used the standard three-equation melt rate parameterization of Holland and Jenkins (1999) to convert the modeled or observed water properties and velocities to an estimated submarine melt rate. There is good agreement between the melt rates estimated with MITgcm and with observations (Slater et al., 2018, their Fig 3). Here, we only consider the modeled melt rates (Fig 5), which have the advantage of covering the

whole extent of the glacier front (unlike rates inferred from observations, which have data gaps in and around the plumes).

There is large spatial variability in submarine melt rates along the glacier front (Fig 5). Submarine melt rates are highest (both in a depth-averaged and maximum sense) within the two plumes where the discharge of buoyant surface meltwater from beneath the glacier gives high water velocities. Outside of the two plumes melt rates are much smaller in a depth-averaged sense, however the lateral circulation excited by the plumes combines with warm surface waters to give high melt rates near the surface outside of the plumes (Fig 5c; Slater et al., 2018).

While these melt-rate estimates represent the state-of-the-art in terms of melt-rate modeling, we stress that they are based on a melt-rate parameterization that has not been confirmed by observations, especially for the case of a mostly vertical front of a tidewater glacier. The uncertainty associated with these melt-rate estimates is further discussed in Section 5."

*2. The calving rate was estimated by (i) locating calving events with measurements of ocean waves and (ii) converting measurements of calving events into calving rates by somehow scaling the number of events (and amplitude of the waves) so as to roughly balance equation 1. Given the wide variety of types of calving events and iceberg geometries and poor understanding of wave generation by calving events, it seems dangerous to assume any relation between wave amplitude and calving event size. Using this method, the authors are unable to close the mass budget (see Fig. 6c) along the southern part of the terminus, which they attribute to the style of calving that they observe there. That's fine, I suppose, but elsewhere along the terminus the left- and right-hand sides of equation 1 differ by a factor of 2 or more, which raises questions about the validity of their results — and in particular the modeled melt rates and the estimated calving rates.*

We thank the reviewer for this feedback. We appreciate that there are large uncertainties associated with the calving term and that this makes it difficult to balance the mass flux at the terminus. In the revised version of the manuscript we have therefore put much of the weight of our argument on the perspective of computing the calving flux as the residual from the other three terms in the flux balance equation (see also next comment).

However, the calving estimates from the pressure sensor data provide insightful independent results regarding the spatial distribution of the frequency and (to a more limited extent) the volume flux due to calving. Overall, the combination of calving data and melt flux data is unique and it provides at least rough estimates for all terms in eq 1. Our results highlight the challenges of balancing the mass flux, even when ocean and calving conditions are known, and emphasize the need to reconcile commonly accepted melt rate estimates with the spatial variability across the front.

We agree that a relation between wave amplitude and calving event size should be adopted with great caution. However, to a first rough approximation, it might be assumed that larger calving events are correlated with larger wave amplitudes. The two main characteristics that can skew this direct correspondence are the geometry of the calving block (is it an efficient wave-generator or not) and the height from which it falls. If certain

regions or times produce calvings that are more efficient wave generators than the average this would lead to an overestimation in calving volume for those regions or times. The same applies to calving heights, as illustrated in the promontory: here we know that the high cliffs produce frequent (but small) calvings from great heights which leads to a spuriously elevated volume flux in this region. For the main part of the glacier, however, it may be a satisfactory first approximation that geometries and heights are roughly evenly distributed so that we can statistically expect that larger calving events are correlated with larger wave amplitudes.

However, such a relationship is indubitably speculative. For the final flux balance estimates we therefore invoked the simpler approximation that for the main part of the glacier higher calving frequencies correspond to higher calving fluxes. This led us to use a scaled frequency of calving events as an estimate for calving volume, thus eschewing the added uncertainties that come with trying to draw conclusions from the wave amplitudes (as pointed out by the reviewer). This is further supported by the result that the calving volume as estimated using wave amplitudes (black line in Fig 5b) is spatially correlated with the observed calving frequency (red bars in Fig 5b), i.e., the size of calving events appears to be fairly uniformly distributed along the glacier front.

We have aimed to emphasize these caveats and clarify the limitations associated with the calving data at several places in the manuscript, and particularly in the following revised paragraph:

"Even though this dataset presents a rather accurate record of calving frequencies, it remains challenging to infer a total volume of calved ice (Minowa et al., 2018). This is due to the different modes of calving (e.g., ice-cliff calving versus submarine calving), as well as the different shapes of calved ice blocks and the differing heights from which they fall (or depths from which they rise). Distinguishing between these events from the pressure sensor data is a difficult task and beyond the scope of this study. The pressure sensors do record an amplitude of the incoming wave packet associated with a given calving event. Crudely approximating that this amplitude is proportional to the size of the calved ice we can estimate a relative calving volume (black curve in Fig 6b). However, since a small cone-shaped ice block can act as a more efficient wave generator than a large flat piece of ice (N. Pizzo, personal communication, Bühler, 2007), it is difficult to ascertain a direct relation between wave amplitudes and calving volume. In what follows we therefore only consider the calving frequency record and will scale this record such that the resulting calving flux approximately closes the mass budget at the glacier front (see Section 4.2). Given the limitations of the data, we take such a scaling to be the most justifiable first-order approximation, supported by the rather uniform distribution of estimated calving event sizes along the glacier front. The scaling factor is chosen such that the mean calving volume is equal to the mean of the residual, i.e. $\langle C \rangle = \langle H(R + vi) - DM \rangle$."

*It's not clear to me why the calving rate wasn't just calculated by subtracting the melt rate from the left-hand side of equation 1, which would at least ensure that mass is being conserved. Nonetheless, a major conclusion in the paper is that frontal ablation is dominated by melting in a couple of specific locations but is otherwise dominated by calving. That is an interesting result, but I think that point can be made much more simply, without the need to convert wave amplitudes to calving event sizes, and may*

*already be apparent in the modeling paper that is in review. For example, in figure 6b it is apparent that the modeled melt fluxes are highly focused on certain regions of the terminus, which already implies that the calving fluxes must be comparatively large everywhere else.*

We thank the reviewer for this comment and have added the suggested calculation to the manuscript: we now provide an estimate of the calving flux (per unit width) as the residual of the other three terms in eq 1. This is shown in the revised Fig 7c. The two methods compare well in terms of magnitude of the calving flux except in the area of the promontory where the elevated cliff heights lead to a different calving regime with frequent smaller calving events from greater heights, as discussed above and in the text.

We maintain that the calving flux as estimated from observations is a valuable addition here, since it provides an independent estimate, and highlights possible issues with the modeled melt-rates: the calving distribution along the glacier front is more homogenous than the extremely localized melt rates would suggest. We conclude from this result that conventional melt rates should be revisited. This is discussed in more detail in the revised manuscript in section 5.1.

*I include some detailed comments below, focusing on areas where I found the descriptions to be somewhat ambiguous. In addition, all errors are discussed in the text, I think it would be helpful to include error bars or confidence intervals in the figures (especially figure 6).*

We thank the reviewer for the comments above as well as the specific comments below, which we have addressed to our best ability.

Error bars have been added to Figures 3 and 7 (formerly Fig 6). Quantifying the uncertainty in the melt rates is somewhat challenging, since there are uncertainties associated with the melt parameterization themselves, as well as with the model and observations. We have expanded our discussion of the uncertainties inherent in the melt rate estimation as follows:

"The finding that calving appears to make up almost the entire loss of ice is somewhat unexpected, in particular since during the study period the glacier's calving activity was limited to relatively small events, and the fjord was by-and-large devoid of icebergs. Furthermore, the melt rates used here are roughly double that of what previous estimates would have been since we account for additional melt that arises from the recirculation of warm ambient surface waters (Slater et al., 2018). Yet, melting supposedly only makes up ~6% of the total ablation.

Given the lack of observational verification of the current melt rate parameterization, it is worth considering end-member melt rate scenarios. Perhaps the key parameter in the melt rate parameterization is the thermal Stanton number, which directly controls the rate of transfer of heat from the ocean to the ice. Its canonical value is based largely on field observations at a cold Antarctic ice shelf and there are as yet no strong observational constraints from tidewater glaciers. Furthermore, Ezhova et al. (2018) have recently argued for a larger Stanton number based on direct numerical simulations. We thus consider lower and upper bounds for melt rates in which the thermal Stanton number is respectively reduced and increased by 50% (Fig 7b,c).

To obtain the upper bound melt rate scenario, we also increase the outside-of-plume water velocity which enters the melt rate parameterization. While vertical velocities inside of plumes might be considered reliable based on well-validated plume theory (Morton et al., 1956), one could argue that the mean modeled outside-of-plume velocities may be too small for a number of reasons including coarse model resolution and the lack of tides, surface waves, and calving events which may excite water motion. These factors might crudely be taken into account by placing an additional velocity in the melt-rate parameterization. Such an approach has some precedent with the inclusion of tides beneath ice shelves (Jenkins and Nicholls, 2010). In the upper bound melt rate scenario, we thus add 0.2 m/s to the outside-of-plume water velocity entering the melt rate parameterization.

In the lower bound melt rate scenario, melting accounts for an even smaller fraction of mass loss than in our best estimate, but is still significant inside the plumes. In the upper bound melt rate estimate, melting accounts for a significant proportion of mass loss both inside and outside of the plumes (Fig 7b). Clearly this is an observationally under-constrained discussion, and we emphasize that these upper and lower bounds are very rough error estimates as the state of understanding of submarine melting does not yet permit rigorous quantitative assessment of uncertainties. Yet these bounds do show that through reasonable modification of the melt rate parameterization, melting can account for a larger fraction of the ice loss than reported in our best estimate. Even by introducing these uncertainties, however, the analysis presented still indicates that calving is the dominant mode of mass loss for most of the glacier front (except at the localized melt plumes)."

**Specific comments:**

**p. 1, l. 14:** *Tidewater glaciers aren't boundaries per se but rather a part of the ice sheet. Also, tidewater glacier termini (and the bottom of the remaining ice shelves) are really the only boundary between the Greenland Ice Sheet and the ocean.*

We thank the reviewer for this comment and have adjusted the text to simply read:

"Tidewater glaciers act as thermodynamic buffers as well as mechanical buttresses between the ocean and the Greenland ice sheet (Rignot and Thomas, 2002; Howat et al., 2007; Nick et al., 2009)."

**p. 2, l. 1-2:** *Its a little confusing how these sentences jump from ablation (in general) to specifically talking about frontal ablation.*

We have added 'frontal' to the first use of 'ablation' for better clarification.

**p. 2, l. 15:** *You could also mention that ocean surface gravity waves (both shortand long-period) have been used to observe calving events in Greenland in previous studies.*

This has been added, together with a reference to Minowa et al (2018).

**p. 2, l. 26:** *It wasn't clear to me how the hydrographic data was used to infer melt /*

*constrain the submarine melt model.*

We thank the reviewer for this comment — this is now discussed in more detail in Section 3.3 (see general comments above).

***p. 4, l. 1:*** *In what way is the plume amplified? The drainage event is relatively shortlived, so I suspect it wouldn't have an impact on the plume over time-scales longer than about a week.*

We thank the reviewer for this comment and have changed the text to read:

"While this plume appears to be an annually recurring feature, its discharge is likely amplified episodically by the cyclical drainage of the large ice-dammed lake Tininnilik located to the south-west of the promontory (Kjeldsen et al 2017)."

***p. 4, l. 28:*** *Did you correct the DEM to account for the difference between the ellipsoid and the geoid?*

Here we set the reference height at the estimated current sea level as measured by the DSM a few hundred meters from the terminus (so as to avoid bleed-in effects from the ice—ocean boundary), and we measure the height of the glacier as simply the difference between the estimated sea surface and the height of the ice at the glacier front (i.e., we only consider relative, not absolute, heights).

***Figure 2, caption:*** *It doesn't really make sense to talk about the flux being into the page here. Do you mean that the velocity profile is from the perspective of somebody looking down glacier?*

We thank the reviewer for this suggestion. The text has been changed to read:
"Here, as in all figures, the orientation is looking down-glacier."

***p. 5, l. 11:*** *Speculation shouldn't be part of the methods section.*

We appreciate the reviewer's point, and have moved this paragraph to later in the text. Note, however, that we did not structure the paper with a strict methods—results—discussion layout (so as to tell the story in a more streamlined way) and the discussion of the potential partial flotation of the terminus remains in the section that estimates the ice advection.

***Figure 3b*** *(and elsewhere): I'm not sure if its correct to refer to this quantity as a flux. Should it be a flux per unit width?*

We thank the reviewer for highlighting this inconsistency — it should indeed be flux per unit width. The text has been changed accordingly.

***p. 7, l. 7:*** *You can be more specific here. For an infinite slap with no sliding and uniform temperature, the depth-averaged velocity is 80% of the surface velocity. The percentage goes up for rapidly sliding glaciers and those that have concentrated deformation at depth, such as tidewater glaciers.*

We thank the reviewer for this suggestion and have adjusted the manuscript to read:

"This assumes plug flow, i.e., that the ice velocity is approximately constant from the surface to the ice—bedrock interface. Note that for a glacier with no sliding and uniform temperature, the depth-averaged velocity is 80% of the surface velocity. For fast-flowing tidewater glaciers with concentrated deformation at depth, such as Saqqarliup, plug flow is therefore considered a good approximation (Meier and Post, 1987)."

**Figure 4c:** *For clarity, considering specifying that negative retreat rates indicate advance. (My personal preference is to plot dL/dt, so that positive indicates advance and negative retreat, but this is fine too.)*

We thank the reviewer and have added the following:
"Here, positive *R* represents glacier retreat and negative *R* glacier advance."

**p. 8, l. 9:** *Don't you mean across-glacier variability?*

We thank the reviewer for pointing out this source of confusion. We have changed the text to read:
"However, there is substantial variability along the glacier front in this cycle."

**p. 9, l. 1-2:** *Do you actually calculate the retreat rate perpendicular to the initial terminus, or is it perpendicular to a straight line fit through the initial terminus?*

Yes, it is indeed perpendicular to the initial terminus, which we felt was a more rigorous estimate since parts of the southern section of the terminus are at an acute angle to the main glacier front and a small retreat in those regions can lead to spuriously large retreat when considering a straight line fit.

**p. 9, l. 8-14** *(and elsewhere in the paper): "rates" and "fluxes" are conflated in several places, which may be confusing. For example, in equation (1), H\*R is not the retreat rate.*

We thank the reviewer for pointing out this accuracy and have strived to be more precise in our uses of the terms "rates" and "fluxes" in the revised version of the manuscript.

**p. 9, l. 25:** *derived calving record by: : : ? Should "by" be deleted? Or is something missing here?*

Yes, "by" has been deleted.

**p. 10, l. 16:** *How do you do this scaling? Through a minimization procedure?*

This is correct. The calving frequencies are scaled such that the resulting mean calving flux per approximately balances the flux over the main part of the glacier. Corresponding text has been added to the manuscript to clarify this point:

"The scaling factor is chosen such that the mean calving volume is equal to the mean of the residual, i.e. $\langle C \rangle = \langle H(R + v_i) - DM \rangle$."

**p. 10, l. 13-15:** *Laboratory experiments by Burton et al. (2012) suggest that the wave amplitude of waves produced by capsizing icebergs depends on the energy released during the capsize event, which scales nonlinearly with iceberg geometry.*

We thank the reviewer for pointing us to that result. The link between iceberg capsizing and calving events is interesting, but it is difficult to ascertain how the nonlinearity for iceberg capsizing translates to calving, and we have therefore chosen to eschew a discussion of this question in the paper.

**p. 10, l. 24:** *I assume that the modeled melt rates are constrained by hydrographic observations. Is that correct?*

This is correct. We have clarified this in the revised manuscript as follows:

"Both buoyant plume theory and MITgcm were forced with runoff from the regional climate model RACMO and initialized with hydrographic profiles from the fjord. Slater et al. (2018) also presented observationally-inferred melt rates using water property and velocity measurements collected within 100 m of the calving front. In each approach, Slater et al. (2018) then used the standard three-equation melt rate parameterization of Holland and Jenkins (1999) to convert the modeled or observed water properties and velocities to an estimated submarine melt rate. There is good agreement between the melt rates estimated with MITgcm and with observations (Slater et al., 2018, their Fig 3). Here, we only consider the modeled melt rates (Fig 5), which have the advantage of covering the whole extent of the glacier front (unlike rates inferred from observations, which have data gaps in and around the plumes)."

**p. 12, l. 21:** *Doesn't the thickness have to go to 0 at the margins?*

We thank the reviewer for raising this point. We have added "Away from the margins, …" to clarify our statement.

**p. 12, l. 29-30:** *Seems self-evident to me.*

We agree with the reviewer and have replaced "which suggests that" with "since".

**p. 14, l. 4-5:** *Can you quantify the amount of ice in the fjord and how it varies seasonally (even in a rough sense by looking at satellite imagery)? This glacier is pretty slow, so the calving fluxes would have to be pretty small.*

We agree with the reviewer that helpful insights may be gained from looking at the volume of ice in the fjord. However, the estimation of seasonal variations is made difficult by the lack of wintertime data and also the presence of sea ice in the fjord for much of the year. We feel that such estimates would go beyond the scope of the present study.

**Reviewer #2**

*Overall I think this is an interesting paper. It's a tricky topic, but an important one, and so it is nice to see people try to improve our knowledge. However, I do have a couple of major concerns about the fundamental approach. The authors are essentially trying to determine the balance between ice losses from calving and those from melting. However, it seems to be that both of these things are poorly constrained:*

*The melt rates are calculated using a model and observations, but these are from Slater et al., which is currently 'submitted'. As such, it is not accessible to me and I also think it is inappropriate to treat this as accepted, when it hasn't been peer-reviewed. I'm not saying this is the case, but if Slater et al were to be rejected or the technique deemed in appropriate, it would mean that this paper also had the same issues.*

We share the reviewer's concerns and are grateful for this feedback. The paper by Slater et al. has since been published at Geophysical Research Letters and is available here: https://agupubs.onlinelibrary.wiley.com/doi/10.1029/2018GL080763.
We have further expanded the discussion of the melt rate estimation such that the present work is a stand-alone unit. Please also see our response to the similar comment of Reviewer #1 on page 1 of this document.

*The other term is calving, for which the authors have frequency (which is a useful dataset) but no volume / area data. They therefore take it as the residual of the left hand side of the equation, minus melting. As noted above, the melt calculations may have quite high errors and the method is actually not published. As such, I think this limits the confidence we can have in the results.*

In the revised version we estimate calving flux both as the residual of the other terms in eq 1 (as also suggested by Reviewer 1) and by using the independent (scaled) measured calving frequencies. This sheds some more light on the uncertainty associated with our calving estimates. Furthermore, we believe that the more robust justification of the melt estimates in the revised manuscript (and the publication of Slater et al., GRL, 2018) will provide better support for our discussion of the calving flux estimates.

*The approach of using waves to ID calving events has been demonstrated in Minowa et al 2018 but they also had time-lapse imagery, which meant they could determine the type of calving (topple etc) and they did not need size. Here, I think the size is needed, so that the calving term is more robust.*

We fully agree that the knowledge of either the type of calving or the size of the calved ice would be required to exactly close the mass flux across the terminus. In fact, we do have simultaneous time-lapse imagery for parts of the calving record. (This imagery is of sufficient resolution to validate the accuracy of the pressure-wave detection, but it is not detailed enough to reveal the exact type of calving for individual events, nor the size of the broken-off pieces.) However, during the field experiments we observed that there are two areas of the glacier with markedly distinct calving regimes - the promontory and the main glacier. Within those two regions the calving events appeared more uniform. This is further supported by the fact that the measured wave amplitudes do not highlight

substantial differences in calving energies along the main part of the glacier. We would therefore argue that a scaled calving frequency — even though an imperfect measure — is a helpful first step toward robustly estimating the calving flux along the glacier face.

*A more minor point is the paper structure: the methods and results are mixed together. I'm not totally against this and I can see why the authors have taken the approach here (as you need to get the results from one part to do the next), but I think it does reduce the accessibility for someone trying to repeat the experiments, who just wants to get at the methods. As noted above, I think it needs to at least briefly describe the methods in Slater et al submitted here (to avoid the reader having to go and find another publication). Slater et al should also be through the peer review process before it is cited, as it's central to the argument.*

We thank the reviewer for the comment regarding the structure and have restructured the paper with the more traditional sequence of methods, results, discussion in mind. In the revised version, we provide two combined methods/results sections that present the physical setting and estimates of the 4 terms of the flux balance (Sections 2 and 3). This is followed by two discussion sections (Sections 4 and 5), which discuss the spatial variability and relative importance of melting and calving. Following the reviewer's suggestion, and in light of the discussion above, we have further expanded the discussion of the melt estimates so as to guarantee this paper to be stand-alone.

*Finally, I think the abstract is a bit misleading over time frames. You have the calving data for July, but the earlier material implies that this is a year-round budget.*

We thank the reviewer for raising this point. We have added "in the summer" to the first sentence of the abstract for clarification.

*I've attached some minor comments in the annotated pdf.*

Comments from the annotated pdf:

**p. 1, l. 5:** *melting-dominated etc.*

This has been fixed.

**p. 1, l. 14:** *Also Jensen et al 2016. I'd add another paper or two as examples.*

We thank the reviewer for this suggestion. A reference to Jensen et al (2016) has been added.

**p. 1, l. 16:** *I'd say since the 2000s, as it wasn't consistent - see e.g. Howat et al., 2008 for east Greenland.*

This has been changed accordingly.

**p. 1, l. 18:** *May want to add Hill EA, Carr JR, Stokes CR. A Review of Recent Changes in Major Marine-Terminating Outlet Glaciers in Northern Greenland. Frontiers in Earth Science 2017, 4, 111.*

We thank the reviewer for this suggestion. A reference to Hill et al (2017) has been added.

*p. 2, l. 13: Might be worth noting that data in fjords are very limited, and the data we have are only from a small number of glaciers (e.g. the Rignot paper on W Greenland, Helheim, Peterman etc) which further impacts our capacity to understand melt rates.*

We thank the reviewer for this suggestion and have adjusted the text to read:
"In-situ ablation data remains scarce, and previous studies of explicit calving activities of Greenland's tidewater glaciers have typically been limited to visible daylight hours […]"

*p. 2, l. 19: I defer to the editor as to whether 'submitted' papers are allowed. Is there not a published paper you can cite here?*

References to Slater et al (submitted), have been changed to Slater et al (2018), as discussed above.

*p. 2, l. 31: Why this glacier? If it's for logistical reasons, there's no issue but I'd just note it.*

We have added the following text for clarification:

"This site was chosen because ocean properties and bathymetry could be measured within 100 m of the ice front. Such observations are exceedingly difficult to obtain at larger glaciers which often have an ice mélange that obstructs access and where calving poses a major threat to equipment and personnel."

*p. 4, l. 1: ... to be an annually recurring….*

This has been fixed.

*p. 4, l. 5: we will. Please avid shortenings like this.*

This has been fixed. We apologize for the sloppiness.

*p. 4, l. 9: Which two? Restate the dates*

This has been changed to 'the 2012 and 2013 field seasons'.

*p. 4, l. 11: Spell out acronyms the first time they are used.*

As requested, we have spelled out the acronyms for REMUS, ADCP, NMEA, and XCTD.

*p. 4, l. 11: these data. 'Data' is a plural noun.*

This has been changed.

***p. 4, l. 20-26:*** *This feels more like results than methods. Please make it clear whether this background info from a previous study, needed to understand the methods, or move it to the appropriate place in the results.*

As discussed above, we have attempted to structure the revised article in a more traditional way.

***p. 5, l. 1:*** *Again, seems like results not methods. I looked back and the title of the section includes the physical setting, but it just seems odd having this here, as these are really results to me. Please revise this throughout this section.*

*Reading further on, there doesn't appear to be a results section. Personally, I'd prefer to see separate methods and results sections, as I think it's easier for the reader to access the information they need.*

Please see our response to the comment above.

***p. 6, l. 9:*** *Terms 'stream' and 'drainage location area bit confusing here - I assume you mean where the fast ice flow meets the ocean, but it could be mis-read as a water stream.*

We thank the reviewer for this request for clarification and have changed the text to read "ice stream".

***p. 7, l. 1:*** *Please indicate which months these are.*

We have added "(June—September)".

***p. 7, l. 2:*** *Why is it not shown?*

Please see response to next comment.

***p. 7, l. 2:*** *So is this why you're excluding it? I.e. because this is an 'extreme' event? Please state this.*

We thank the reviewer for highlighting this, and have adjusted the text to read:

>  "This slowdown is not shown here as it has been linked to a major drainage event of lake Tininnilik (Kjeldsen et al, 2017), and therefore subject to altogether different environmental forcing."

***Figure 5b:*** *This is really interesting information, but the figure is so small I struggle to read it. Please make it bigger so that the dots are easier to see.*

The figure has been revised such that the original sub-figure 5a is now an inset to the former Fig 5b, which in turn is substantially larger now. (This is figure 6 in the revised manuscript).

*p. 8, l. 7: seasonal cycle in terminus position, with a mean advance for the entire front of…..*

We thank the reviewer for this suggestion. The text has been changed accordingly.

*p. 9, l. 20: How Representative do you think this is of the rest of the summer? I think you should make clearer in your abstract that this is a summer balance - if you only have calving events for this period, it can't be a year round balance. It would also be helpful to give this context e.g. do 50% of all calving events in a given year occur in this time or is it 90%?*

We thank the reviewer for this comment and have changed the beginning of the abstract to clarify that we are looking at the summer months only. It is difficult to estimate how much of a year's calving activity occurs during the summer, since there is close to no data of wintertime conditions. We believe that such an estimate is beyond the scope of this study.

*p. 10, l. 1: This is an interesting point, as I would expect the calving frequency to be highest where the glacier is thickest (and deepest bed).*

We agree with the reviewer.

*p. 10, l. 29: Are these your results or Slaters? If the former, please show them. Might be helpful to add the figure either way, so the reader can easily refer to it.*

A melt figure, using data from Slater et al (2018), has been added (Fig 5).

*p. 15, l. 29: This has been observed on a larger scale for lake-terminating glaciers in New Zealand. E.g. Robertson et al 2013.*

We thank the reviewer for this suggestion and have added references to Robertson et al (2012), as well as the review by Benn et al (2017).

[revised manuscript text omitted]